# *Swap and Predict* – Predicting the Semantic Changes in Words across Corpora by Context Swapping

**Taichi Aida**
Tokyo Metropolitan University
aida-taichi@ed.tmu.ac.jp

**Danushka Bollegala**
Amazon, University of Liverpool
danushka@liverpool.ac.uk

## Abstract

Meanings of words change over time and across domains. Detecting the semantic changes of words is an important task for various NLP applications that must make time-sensitive predictions. We consider the problem of predicting whether a given target word, $w$, changes its meaning between two different text corpora, $\mathcal{C}_1$ and $\mathcal{C}_2$. For this purpose, we propose *Swapping-based Semantic Change Detection* (SSCD), an unsupervised method that randomly swaps contexts between $\mathcal{C}_1$ and $\mathcal{C}_2$ where $w$ occurs. We then look at the distribution of contextualised word embeddings of $w$, obtained from a pretrained masked language model (MLM), representing the meaning of $w$ in its occurrence contexts in $\mathcal{C}_1$ and $\mathcal{C}_2$. Intuitively, if the meaning of $w$ does not change between $\mathcal{C}_1$ and $\mathcal{C}_2$, we would expect the distributions of contextualised word embeddings of $w$ to remain the same before and after this random swapping process. Despite its simplicity, we demonstrate that even by using pretrained MLMs without any fine-tuning, our proposed context swapping method accurately predicts the semantic changes of words in four languages (English, German, Swedish, and Latin) and across different time spans (over 50 years and about five years). Moreover, our method achieves significant performance improvements compared to strong baselines for the English semantic change prediction task.[1]

## 1 Introduction

Meaning of a word is a dynamic concept that evolves over time (Tahmasebi et al., 2021). For example, the meaning of the word *gay* has transformed from *happy* to *homosexual*, whereas *cell* has included *cell phone* to its previous meanings of *prison* and *biology*. Automatic detection of words whose meanings change over time has provided important insights for diverse fields such as

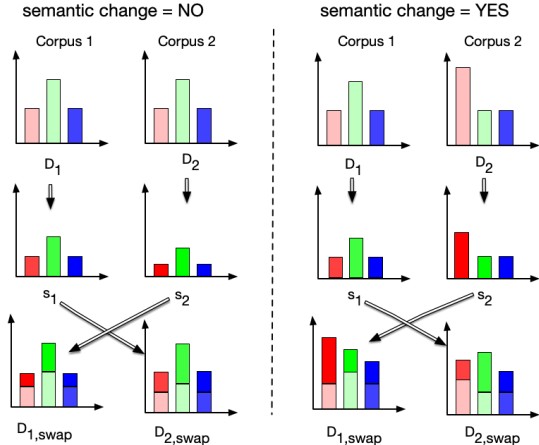

Figure 1: Overview of SSCD. **Left:** The contextualised word embedding distributions, $D_1$ and $D_2$ of a word which has not changed its meaning between the two corpora. Two random samples of equal number of sentences containing the target word, $s_1$ and $s_2$, are taken respectively from $D_1$ and $D_2$ and swapped between the corpora. Here, we see that the contextualised word embedding distributions after swapping (i.e. $\mathcal{D}_{1,\text{swap}}$ and $\mathcal{D}_{2,\text{swap}}$) are similar to those before, thus preserving the distance between distributions. **Right:** For a word that has different meanings in the two corpora, swapping process pushes both distributions to become similar, thus reducing the distance between the swapped versions smaller to that between the original ones.

linguistics, lexicography, sociology, and information retrieval (Traugott and Dasher, 2001; Cook and Stevenson, 2010; Michel et al., 2011; Kutuzov et al., 2018). For example, in e-commerce, a user might use the same keyword (e.g. *scarf*) to search for different types of products in different seasons (e.g. *silk scarves in spring* vs. *woollen scarves in winter*). The performance of pretrained Large Language Models (LLMs) is shown to decline over time (Loureiro et al., 2022; Lazaridou et al., 2021) because they are trained on static snapshots. If we can detect which words have their meanings changed, we can efficiently fine-tune LLMs to reflect only those changes (yu Su et al., 2022).

---

[1] Source code is available at https://github.com/a1da4/svp-swap .

Detecting whether a word has its meaning changed between two given corpora, sampled at different points in time, is a challenging task due to several reasons. First, a single (polysemous) word can take different meanings in different contexts even within the same corpus. Therefore, creating a representation for the meaning of a word across an entire corpus is a challenging task compared to that in a single sentence or a document. Prior work have averaged static (Kim et al., 2014; Kulkarni et al., 2015; Hamilton et al., 2016; Yao et al., 2018; Dubossarsky et al., 2019; Aida et al., 2021) or contextualised (Martinc et al., 2020; Beck, 2020; Kutuzov and Giulianelli, 2020; Rosin et al., 2022; Rosin and Radinsky, 2022) word embeddings for this purpose, which is suboptimal because averaging conflates multiple meanings of a word into a single vector. Second, large corpora or word lists labelled for semantic changes of words do not exist, thus requiring semantic change detection (SCD) to be approached as an unsupervised task (Schlechtweg et al., 2020).

To address the above-mentioned challenges, we propose **Swapping-based Semantic Change Detection** (SSCD). To explain SSCD further, let us assume that we are interested in detecting whether a target word $w$ has changed its meaning from a corpus $\mathcal{C}_1$ to another corpus $\mathcal{C}_2$. First, to represent the meaning of $w$ in a corpus, SSCD uses the set of contextualised word embeddings (aka. *sibling* embeddings) of $w$ in all of its contexts in the corpus, computed using a pre-trained Masked Language Model (MLM). According to the distributional hypothesis (Harris, 1954), if $w$ has not changed its meaning from $\mathcal{C}_1$ to $\mathcal{C}_2$, $w$ will be represented by similar distributions in both $\mathcal{C}_1$ and $\mathcal{C}_2$. Prior work has shown that contextualised word embedding of a word encodes word-sense related information that is useful for representing the meaning of the word in its occurring context (Zhou and Bollegala, 2021; Loureiro et al., 2022). Unlike the point estimates made in prior work (Liu et al., 2021) by averaging word embeddings across a corpus (thus conflating different meanings), we follow Aida and Bollegala (2023) and represent a word by a multivariate Gaussian distribution that captures both mean and variance of the sibling distribution. Various distance/divergence measures can then be used to measure the distance between the two sibling distributions of $w$, computed independently from $\mathcal{C}_1$ and $\mathcal{C}_2$.

An important limitation of the above-described approach is that it **provides only a single estimate** of the semantic change from a given pair of corpora, which is likely to be unreliable. This is especially problematic when the corpora are small and noisy. To overcome this limitation, SSCD randomly samples sentences $s_1$ and $s_2$ that contain $w$, respectively from $\mathcal{C}_1$ and $\mathcal{C}_2$ and swaps the two samples between $\mathcal{C}_1$ and $\mathcal{C}_2$. The intuition behind this swapping step is visually explained in Figure 1. On average, a random sample of distribution will be similar to the sampling distribution (Dekking et al., 2005). Therefore, if $w$ has similar meanings in $\mathcal{C}_1$ and $\mathcal{C}_2$, the distance between sibling distributions before and after the swapping step will be similar. On the other hand, if $w$'s meaning has changed between the corpora, the distributions after swapping will be different from the original ones, thus having different distances between them. SSCD conducts this sampling and swapping process multiple times to obtain a more reliable estimate of the semantic changes for a target word.

We evaluate SSCD against previously proposed SCD methods on two datasets: SemEval-2020 Task 1 (Schlechtweg et al., 2020) and Liverpool FC (Del Tredici et al., 2019), which cover four languages (English, German, Swedish, and Latin) and two time periods (some spanning longer than fifty years to as less than ten years). Experimental results show that SSCD achieved significant performance improvements compared to strong baselines. Moreover, SSCD outperforms the permutation test proposed by Liu et al. (2021) on both datasets and in three languages, showing the generalisability of SSCD across datasets and languages. Moreover, we show that there exists a trade-off between the percentage of sentences that can be swapped between two corpora (i.e. swap rate) and the performance of SSCD, and propose a fully unsupervised method that does not require labelled data to determine the optimal swap rate.

## 2 Related Work

The phenomenon of diachronic semantic change of words has been extensively studied in linguistics (Traugott and Dasher, 2001) but has recently attracted interest in the NLP community as well. **Unsupervised SCD** is mainly conducted using word embeddings, and various methods have been proposed that use static word embeddings such as initialisation (Kim et al., 2014), alignment (Kulkarni

et al., 2015; Hamilton et al., 2016), and joint learning (Yao et al., 2018; Dubossarsky et al., 2019; Aida et al., 2021). In recent years, with the advent of pretrained MLMs, word embeddings can be obtained per context rather than per corpus, and the set of contextualised word embeddings (sibling embeddings) can be used for the semantic change analysis (Hu et al., 2019; Giulianelli et al., 2020) and detection (Martinc et al., 2020; Beck, 2020; Kutuzov and Giulianelli, 2020; Rosin et al., 2022; Rosin and Radinsky, 2022; Aida and Bollegala, 2023; Cassotti et al., 2023).

Recent research has mainly focused on MLMs, which embed useful information related to the meaning of words in their embeddings (Zhou and Bollegala, 2021). While prior work use pretrained or fine-tuned MLMs without considering temporal information, Rosin et al. (2022) proposed a fine-tuning method by adding temporal tokens (such as <2023>) at the beginning of a sentence. In the fine-tuning step, MLMs optimise two masked language modelling objectives: 1) predicting the masked time tokens from given contexts, and 2) predicting the masked tokens from given contexts with time tokens. This method has been shown to outperform the previously unbeatable static word embeddings in the SCD benchmark, SemEval 2020 Task 1 (Schlechtweg et al., 2020). Moreover, Rosin and Radinsky (2022) proposed adding a time-aware attention mechanism within the MLMs. In the training step, they conduct additional training on the MLM and the temporal attention mechanism on the target data. This model also achieves significant performance improvements in unsupervised SCD benchmarks.

Yu Su et al. (2022) applied SCD to the temporal generalisation of pretrained MLMs. Pretrained MLMs perform worse the further away in time from the trained timestamps, and require additional training (Lazaridou et al., 2021; Loureiro et al., 2022). They show that an SCD method can effectively improve the performance of pretrained MLMs, because the additional training can be limited to those words whose meanings change over time. Aida and Bollegala (2023) introduced a method that represents sibling embeddings at each time period using multivariate Gaussians, thus enabling various divergence and distance metrics to be used to compute semantic change scores. Experimental results show that instead of using only the mean of the sibling embeddings, it is important

to consider also its variance, to obtain performance comparable to the previous SoTA. Consequently, we use sibling embeddings to represent the distribution of a word in a corpus in SSCD.

However, existing methods make predictions only once for a given target word. Such point estimates of semantic change scores are unreliable especially when a target word is rare in a corpus. To overcome this unreliability, Liu et al. (2021) use context swapping to make multiple predictions for the semantic change of a target word. First, the degree of semantic change is calculated by the cosine distance of the average sibling embeddings between time periods, and the reliability of the prediction is validated by context swapping-based tests (permutation test and false discovery rate). Next, words with low reliability ($p \geqq 0.005$) are excluded from the prediction results, and only the words with high reliability are evaluated. However, this means that it is not possible to assess the semantic change of less reliable words, especially those that are less frequent. In particular, it is desirable to be able to utilise pretrained models without additional training and still be able to detect semantic changes appropriately. As discussed later in § 4.1, SSCD successfully overcomes those limitations in Liu et al. (2021), and consistently outperforms the latter in multiple SCD tasks.

Recently, a **supervised SCD** method called XL-LEXEME (Cassotti et al., 2023), which fine-tunes sentence embeddings produced from an MLM on the Word-in-Context (WiC) (Pilehvar and Camacho-Collados, 2019) dataset has achieved state-of-the-art (SoTA) performance for SCD. During training, they fine-tune the pretrained MLM to minimise the contrastive loss calculated from the sentence pairs in WiC instances such that the distinct senses of a target word can be correctly discriminated. However, it has been shown that these external data-dependent methods are difficult to apply to languages that are not presented in the fine-tuning data (e.g. Latin). Our focus in this paper is on *unsupsevised* SCD methods that do not require such sense-labelled resources. Therefore, we do not consider such supervised SCD methods.

# 3 Swapping-based Semantic Change Detection

## 3.1 Definition

**Problem setting.** Given two text corpora $\mathcal{C}_1$ and $\mathcal{C}_2$, we would like to predict whether a target word

**Algorithm 1** Context Swapping

**Input:** target word $w$, swap rate $r$, sentences in which the target word appears $\mathcal{S}_1^w, \mathcal{S}_2^w$
**Output:** swapped sentences $\mathcal{S}_{1,\text{swap}}^w, \mathcal{S}_{2,\text{swap}}^w$
1: $N_1^w \leftarrow len(\mathcal{S}_1^w)$, $N_2^w \leftarrow len(\mathcal{S}_2^w)$
2: $N_{\text{swap}}^w \leftarrow \min(rN_1^w, rN_2^w)$
3: $\mathbf{s}_1^w \leftarrow \text{random\_sample}(\mathcal{S}_1^w, N_{\text{swap}}^w)$
4: $\mathbf{s}_2^w \leftarrow \text{random\_sample}(\mathcal{S}_2^w, N_{\text{swap}}^w)$
5: $\mathcal{S}_{1,\text{swap}}^w \leftarrow (\mathcal{S}_1^w \setminus \mathbf{s}_1^w) \cup \mathbf{s}_2^w$
6: $\mathcal{S}_{2,\text{swap}}^w \leftarrow (\mathcal{S}_2^w \setminus \mathbf{s}_2^w) \cup \mathbf{s}_1^w$
7: **return** $\mathcal{S}_{1,\text{swap}}^w, \mathcal{S}_{2,\text{swap}}^w$

---

**Algorithm 2** SSCD

**Input:** target word $w$, swap rate $r$, time-specific corpora $\mathcal{C}_1, \mathcal{C}_2$, masked language model $M$, divergence/distance function $d$
**Output:** semantic change score
1: $\mathcal{S}_1^w \leftarrow \text{obtain\_sentences}(w, \mathcal{C}_1)$
2: $\mathcal{S}_2^w \leftarrow \text{obtain\_sentences}(w, \mathcal{C}_2)$
3: $\mathcal{D}_1^w \leftarrow \{M(w, s) | s \in \mathcal{S}_1^w\}$
4: $\mathcal{D}_2^w \leftarrow \{M(w, s) | s \in \mathcal{S}_2^w\}$
5: $e_{\text{original}} \leftarrow d(\mathcal{D}_1^w, \mathcal{D}_2^w)$
6: obtain swapped sentences $\mathcal{S}_{1,\text{swap}}^w, \mathcal{S}_{2,\text{swap}}^w$ from Algorithm 1
7: $\mathcal{D}_{1,\text{swap}}^w \leftarrow \{M(w, s) | s \in \mathcal{S}_{1,\text{swap}}^w\}$
8: $\mathcal{D}_{2,\text{swap}}^w \leftarrow \{M(w, s) | s \in \mathcal{S}_{2,\text{swap}}^w\}$
9: $e_{\text{swap}} \leftarrow d(\mathcal{D}_{1,\text{swap}}^w, \mathcal{D}_{2,\text{swap}}^w)$
10: **return** $|e_{\text{original}} - e_{\text{swap}}|$

---

$w$ has its meaning changed from $\mathcal{C}_1$ to $\mathcal{C}_2$. Here, we assume that $\mathcal{C}_1$ is sampled from an earlier time period than $\mathcal{C}_2$. Let the set of sentences containing $w$ in $\mathcal{C}_1$ be $\mathcal{S}_1^w$ and that in $\mathcal{C}_2$ be $\mathcal{S}_2^w$. According to the distributional hypothesis, if the distribution of words that co-occur with $w$ in $\mathcal{S}_1^w$ and that in $\mathcal{S}_2^w$ are different, we predict $w$ to have its meaning changed from $\mathcal{C}_1$ to $\mathcal{C}_2$.

**Meaning modelling.** Given a sentence $s$ and a target word $w$ that occurs in $s$, we obtain the contextualised token embedding $M(w, s)$ of $w$ in $s$, produced by an MLM $M$. We refer to $M(w, s)$ as a *sibling embedding* of $w$. Given a set $\mathcal{S}^w$ of sentences containing $w$, we define the *set of sibling embeddings* by $\mathcal{D}^w = \{M(w, s) \mid s \in \mathcal{S}^w\}$. Let the mean and the covariance of $\mathcal{D}^w$ be respectively $\boldsymbol{\mu}^w$ and $\boldsymbol{\Sigma}^w$.[2]

Based on prior work that show contextual word embeddings to encode useful information that determine the meaning of a target word (Zhou and Bollegala, 2021), we form the following working assumption.

> **Assumption.** *The set of sibling embeddings $\mathcal{D}^w$ represents the meaning of a word $w$ in a corpus $\mathcal{C}$.*

We can reformulate this assumption as a null hypothesis – *the meaning of $w$ has not changed from $\mathcal{C}_1$ to $\mathcal{C}_2$, (unless the corresponding sibling distributions have changed)*, for the purpose of computing the probability that $w$ has its meaning changed from $\mathcal{C}_1$ to $\mathcal{C}_2$.

To validate this hypothesis, SSCD uses the context swapping process described in Algorithm 1. We first find the set of sentences that contain $w$

---

[2]Following Aida and Bollegala (2023), we use diagonal covariance matrices, which are shown to be numerically more stable and accurate for approximating sibling distributions.

---

in $\mathcal{C}_1$ and $\mathcal{C}_2$, denoted respectively by $\mathcal{S}_1^w$ and $\mathcal{S}_2^w$. We then randomly select subsets $\mathbf{s}_1^w \in \mathcal{S}_1^w$ and $\mathbf{s}_2^w \in \mathcal{S}_2^w$, containing exactly $N_{\text{swap}}^w$ number of sentences, determined by the number of sentences in the smaller set between $\mathcal{S}_1^w$ and $\mathcal{S}_1^w$, and the swap rate $r(\in [0, 1])$ (Lines 2-4). Next, we exchange $\mathbf{s}_1^w$ and $\mathbf{s}_2^w$ between $\mathcal{C}_1$ and $\mathcal{C}_2$ and obtain swapped sets of sentences $\mathcal{S}_{1,\text{swap}}^w$ and $\mathcal{S}_{2,\text{swap}}^w$ (Lines 5 and 6). The context swapped corpora are next used to compute a semantic change score for $w$ by SSCD as described in Algorithm 2.

If the sets of the mean and the covariance are *different* between the two time periods (i.e. $\boldsymbol{\mu}_1^w \neq \boldsymbol{\mu}_2^w$ and $\boldsymbol{\Sigma}_1^w \neq \boldsymbol{\Sigma}_2^w$), a non-zero distance will remain between the two distributions (i.e. $e_{\text{original}} > 0$). In this case, the context swapping process described in Algorithm 1 will produce sibling distributions $\mathcal{D}_{1,\text{swap}}^w$ and $\mathcal{D}_{2,\text{swap}}^w$ that are *different* to the corresponding original (i.e. prior to swapping) distributions $\mathcal{D}_1^w$ and $\mathcal{D}_2^w$. Therefore, the distance $e_{\text{swap}}$ between $\mathcal{D}_{1,\text{swap}}^w$ and $\mathcal{D}_{2,\text{swap}}^w$ will be different from $e_{\text{original}}$ between $\mathcal{D}_1^w$ and $\mathcal{D}_2^w$, producing a large $|e_{\text{original}} - e_{\text{swap}}|$. SSCD repeatedly computes $e_{\text{swap}}$ using multiple random samples (20 repetitions are used in the experiments) to obtain a reliable estimate for $|e_{\text{original}} - e_{\text{swap}}|$. Therefore, the null hypothesis could be rejected with high probability, concluding that $w$ to have changed its meaning between $\mathcal{C}_1$ and $\mathcal{C}_2$.

On the other hand, if the sets of the mean and the covariance remain *similar* between the two target time periods (i.e. $\boldsymbol{\mu}_1^w \approx \boldsymbol{\mu}_2^w$ and $\boldsymbol{\Sigma}_1^w \approx \boldsymbol{\Sigma}_2^w$), the original semantic distance $e_{\text{original}}$ (Line 5 in

Algorithm 2) will be close to zero. Moreover, the distance between $e_{\text{original}}$ and $e_{\text{swap}}$ (Line 10) will also be close to zero because swapping would not change the shape of the sibling distributions (i.e. $e_{\text{original}} \approx e_{\text{swap}}$). Therefore, $|e_{\text{original}} - e_{\text{swap}}|$ will be smaller in this case, hence the null hypothesis cannot be rejected for $w$, and SSCD will predict $w$ to be semantically invariant between $\mathcal{C}_1$ and $\mathcal{C}_2$.

Similar to Liu et al. (2021), SSCD can also be extended to calculate the confidence interval for rejecting the null hypothesis using boostrapping (Efron and Tibshirani, 1994; Berg-Kirkpatrick et al., 2012). This is particularly useful for the binary classification subtask in the SemEval 2020 Task 1 for unsupervised SCD, where we must classify a given word as to whether its meaning has changed between the two given corpora. However, for our evaluation on the ranking subtask we require only the semantic change score.

## 3.2 Metrics

Prior work has shown that the metrics used for predictions are also important and the best performance for different languages is reported by different metrics (Kutuzov and Giulianelli, 2020; Aida and Bollegala, 2023). Our proposed SSCD can be applied with various divergence/distance metrics (Lines 5 and 9 in Algorithm 2) as we describe next.

**Divergence measures:** We use Kullback-Leibler (KL) divergence and Jeffrey's divergence.[3] Following previous work (Aida and Bollegala, 2023), we approximate $\mathcal{D}^w$ by a Gaussian $\mathcal{N}(\boldsymbol{\mu}^w, \boldsymbol{\Sigma}^w)$. Based on this setting, two divergence functions have closed-form formulae, computed from the mean and variance of the two multivariate Gaussians $\mathcal{N}(\boldsymbol{\mu}_1^w, \boldsymbol{\Sigma}_1^w)$ and $\mathcal{N}(\boldsymbol{\mu}_2^w, \boldsymbol{\Sigma}_2^w)$. Since the KL divergence is asymmetric and Jeffrey's divergence is symmetric, we calculate two versions for the KL divergence ($\text{KL}(\mathcal{C}_1||\mathcal{C}_2)$ and $\text{KL}(\mathcal{C}_2||\mathcal{C}_1)$), and one for the Jeffrey's divergence ($\text{Jeff}(\mathcal{C}_1||\mathcal{C}_2)$).

**Distance function:** We use seven distance metrics as follows: Bray-Curtis, Canberra, Chebyshev, City Block, Correlation, Cosine, and Euclidean.[4] In this setting, we calculate the distance between the two mean vectors $\boldsymbol{\mu}_1^w$ and $\boldsymbol{\mu}_2^w$ from $\mathcal{D}_1^w$ and $\mathcal{D}_2^w$, respectively.

**DSCD:** We use the Distribution-based Semantic Change Detection (DSCD) method proposed

by Aida and Bollegala (2023) to measure the distance between two sibling distributions. We randomly sample equal number of target word vectors from the two Gaussians $\mathcal{N}(\boldsymbol{\mu}_1^w, \boldsymbol{\Sigma}_1^w)$ and $\mathcal{N}(\boldsymbol{\mu}_2^w, \boldsymbol{\Sigma}_2^w)$, and calculate the average pairwise distance among those vectors. We use the seven distance metrics mentioned above for this purpose.

## 4 Experiments

### 4.1 Effectiveness of Context Swapping

In this section, we evaluate the effectiveness of SSCD by comparing it against the method that uses context swapping for the reliability of prediction proposed by Liu et al. (2021). The concern with their method is that not all words will be evaluated, as already discussed in § 2. Moreover, they require fine-tuning MLMs, which can be computationally costly for large corpora and for every target language of interest. In this experiment, we show that SSCD successfully overcomes those issues as follows: 1) SSCD uses context swapping for calculating the degree of semantic change and makes appropriate predictions for all words; 2) SSCD uses pretrained multilingual BERT[5] without additional architectural modifications such as temporal attention nor fine-tuning. Following previous experiments (Liu et al., 2021) and findings (Laicher et al., 2021), we obtain token embeddings from the last four layers of the MLM. We use the metrics described in § 3.2 for computing distances.

To evaluate the performance of SCD methods, we use two benchmark datasets: SemEval-2020 Task 1 (Schlechtweg et al., 2020) and Liverpool FC (Del Tredici et al., 2019), which cover four languages (English, German, Swedish, and Latin) for longer (spanning over 50 years) and shorter (spanning less than ten years) time periods. In both datasets, a method under evaluation is required to rank a given word list according to the degree of semantic change, which is subsequently compared against human-assigned ranks using the Spearman rank correlation coefficient ($\rho \in [-1, 1]$), where higher values indicate better agreement with the human ratings.

Results of SSCD against the method proposed by Liu et al. (2021) are shown in Table 1. Due to space limitations, we report SSCD results in the setting with the highest average $\rho$ values taken

---

[3]Definitions are provided in the Appendix A.
[4]Definitions are provided in Appendix B.

[5]We use bert-base-multilingual-cased model published on Hugging Face https://huggingface.co/bert-base-multilingual-cased .

| Model | SemEval | | | | Liverpool FC |
| | English | German | Swedish | Latin | English |
|---|---|---|---|---|---|
| Liu et al. (2021) | | | | | |
| $MLM_{tuned}$ | 0.331 | 0.302 | 0.141 | **0.512** | 0.536 |
| + Permutation Test | 0.341 | 0.304 | 0.162 | 0.502 | **0.561** |
| + False Discovery Rate | 0.339 | 0.304 | 0.162 | 0.502 | 0.478 |
| SSCD | | | | | |
| $MLM_{pre}$, Divergence | 0.209 | 0.547 | 0.127 | 0.460 | 0.470 |
| $MLM_{pre}$, Distance (mean only) | **0.383** | **0.597** | **0.234** | 0.433 | 0.492 |
| $MLM_{pre}$, Distance (DSCD) | 0.364 | 0.476 | 0.199 | 0.410 | 0.364 |
| Cassotti et al. (2023) | | | | | |
| XL-LEXEME (supervised) | 0.757 | 0.877 | 0.754 | -0.056 | N/A |

Table 1: Comparison against prior work in SemEval-2020 Task 1 and Liverpool FC. DSCD indicates the average distance calculated on vectors sampled from the distributions of time-specific sibling embeddings (Aida and Bollegala, 2023).

over 20 rounds for each metric.[6] These results reveal that even with the pretrained MLM (no fine-tuning), SSCD outperforms the SCD method proposed by Liu et al. (2021), which uses fine-tuned MLMs, on three out of the five datasets. It can be seen that SSCD accurately detects the semantic changes of words in four languages (English, German, Swedish, and Latin).

While the above methods are unsupervised, we also include in Table 1 the results of the supervised SCD method, XL-LEXEME (Cassotti et al., 2023). XL-LEXEME achieves SoTA performance in English and German, which are included in the fine-tuning data. Moreover, this model also achieves SoTA in Swedish which does not exist in the labelled data, but Danish, quite similar to Swedish is included. On the other hand, this model performs significantly worse in Latin which is not included in the sense-labelled data. This indicates the difficulty of applying supervised SCD methods in languages that are not in the fine-tuning data.

## 4.2 Comparison against Strong Baselines

Building upon the results of the previous section, we compare SSCD with multiple strong baselines. Here, we take $MLM_{temp}$, a very powerful unsupervised model for SCD (Rosin et al., 2022), as a starting point. $MLM_{temp}$ is the fine-tuned version of the published pretrained BERT-base models[7] us-

ing time tokens (Rosin et al., 2022). They add a time token (such as <2023>) to the beginning of a sentence. In the fine-tuning step, the models use two types of MLM objectives: 1) predicting the masked time tokens from given contexts, and 2) predicting the masked tokens from given contexts with time tokens.

**Cosine**: Rosin et al. (2022) make predictions with the average distance of the target token probabilities or the cosine distance of the average sibling embeddings. According to their results, the cosine distance achieves better performance than the average $\ell_1$ distance between the probability distributions.

**APD**: Kutuzov and Giulianelli (2020) report that the average pairwise cosine distance outperforms the cosine distance. Based on this insight, Aida and Bollegala (2023) evaluate the performance of $MLM_{temp}$ with the average pairwise cosine distance.

**DSCD**: Aida and Bollegala (2023) proposed a distribution-based SCD, considering the distributions of the sibling embeddings (*sibling distribution*). During prediction, they sample an equal number of target word vectors from the sibling distribution (approximated by Gaussians) for each time

---

[6]Full results are shown in § D.1.

[7]Although previous research has shown that reducing the model size down to BERT-tiny improves performance (Rosin

and Radinsky, 2022), the results are only for English and have not been verified in the other languages. Hence, experiments will be conducted using the BERT-base model, which is widely used.

period and calculate the average distance. They report that Chebyshev distance measure achieves the best performance.

**Temp. Att.**: Rosin and Radinsky (2022) proposed a temporal attention mechanism, where they add a trainable temporal attention matrix to the pretrained BERT models. Because their two proposed methods (fine-tuning with time tokens and temporal attention) are independent, they proposed to use them simultaneously. Subsequently, additional training is performed on the target corpus. They use the cosine distance following their earlier work (Rosin et al., 2022).

In our evaluations, we will compare using the results published in the original papers, without re-running them. For SSCD, we use the fine-tuned BERT model $\mathbf{MLM}_{temp}$ in line with previous work (Rosin and Radinsky, 2022; Aida and Bollegala, 2023) and make predictions using the same three types of metrics described in § 4.1. We use the SemEval-2020 Task 1 English benchmark for this evaluation.

**Prediction metrics within SSCD.** Before comparing the performance with strong baselines, we employ various divergence/distance functions presented in § 3.2 in our SSCD and compare their performance. Results within our method are shown in Table 2.[8] Table 2 shows that divergence measures perform better than the distance-based metrics.[9]

**Unsupervised hyperparameter search.** Swap rate is the only hyperparameter in SSCD, which must be specified in both Algorithms 1 and 2. However, benchmarks for SCD tasks do not have dedicated development sets, which is problematic for hyperparameter tuning. To address this problem, we propose an unsupervised method to determine the optimal swap rate as follows. Recall that context swapping is minimising the distance between the sibling distributions computed independently for $w$ from the corpora. Therefore, we consider $e_{\text{swap}}$ (computed in Line 9 in Algorithm 2) as an objective function for selecting the swap rate. Specifically, we plot $e_{\text{swap}}$ against the swap rate in Figure 2 and find the swap rate $\hat{r}$ that minimises $e_{\text{swap}}$. We see that $e_{\text{swap}}$ is minimised at swap rate of 0.6 for all three divergence functions (all three curves are

---

[8]Full results are shown in § D.2

[9]Although we cannot list the standard deviation due to space limitations here, the mean and standard deviation (taken over 20 runs) are shown in § D.2.

| Metric | Spearman | Swap rate Opt. | Est. |
|---|---|---|---|
| **Divergence functions** | | | |
| $\text{KL}(C_1 \| C_2)$ | **0.552** | 0.4 | 0.6 |
| $\text{KL}(C_2 \| C_1)$ | 0.516 | 0.4 | 0.6 |
| $\text{Jeff}(C_1 \| C_2)$ | 0.534 | 0.4 | 0.6 |
| **Distance functions** | | | |
| Bray-Curtis | 0.423 | 0.4 | 0.6 |
| Canberra | 0.345 | 0.4 | 0.6 |
| Chebyshev | 0.372 | 0.4 | 0.6 |
| City Block | 0.443 | 0.4 | 0.6 |
| Correlation | 0.471 | 0.4 | 0.6 |
| Cosine | 0.471 | 0.4 | 0.6 |
| Euclidean | 0.453 | 0.4 | 0.6 |

Table 2: Best performance (measured using $\rho$) obtained using SSCD with different divergence/distance measures. The swap rate at which the best performance is obtained (Swap Opt.) and the optimal swap rate estimated using the unsupervised method (Swap Est.) are shown. All results are averaged over 20 runs. We see that Swap Opt. and Swap Est. are similar and are independent of the measure being used.

overlapping for the most part). The optimal swap rates for all metrics are shown in Table 2 and are at or around 0.4, which are sufficiently closer to the $\hat{r}$ estimated by minimising $e_{\text{swap}}$ for those metrics.

**Context Sampling Strategies for SSCD.** The version of SSCD presented in Algorithm 1 randomly selected (in Lines 3 and 4) contexts for swapping, we refer to as SSCD$_{rand}$. However, besides random sampling, different criteria can be considered when selecting the subsets of contexts for swapping. Considering that a word $w$ would be considered to have its meaning changed, if the furthest meanings of $w$ in $\mathcal{C}_1$ and $\mathcal{C}_2$ are dissimilar, we propose a distance-based deterministic context selection method. Specifically, we sort each sibling embedding of $w$ in $\mathcal{S}_1^w$, in descending order of the distance to the centroid of the sibling embeddings in $\mathcal{S}_2^w$. We then select the top (i.e. furthest) $N_{\text{swap}}^w$ number of sibling embeddings from $\mathcal{S}_1^w$ as $\mathbf{s}_1^w$, as the candidates for swapping in Algorithm 1. Likewise, we select $\mathbf{s}_2^w$ considering the centroid of $\mathcal{S}_1^w$. Let us denote this version of SSCD by SSCD$_{dist}$.

As shown in Table 3, SSCD$_{dist}$ when used with DSCD as the distance metric, obtains a $\rho$ value of 0.563, clearly outperforming all other metrics used

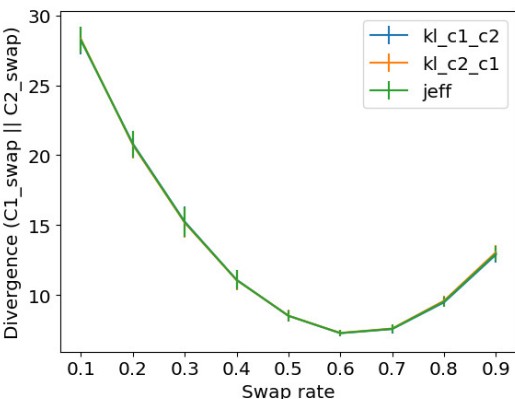

Figure 2: Average divergence of the two distributions after context swapping in all target words. Each plot and error bar shows the mean and standard deviation in 20 seeds, respectively.

with random sampling-based SSCD$_{rand}$.[10]

**Main comparison.** The main result are shown in Table 3 from which we see that both SSCD$_{rand}$ and SSCD$_{dist}$ outperform all other the strong baselines. Moreover, although both the previous best method (Temp. Att.) and our SSCD use the same fine-tuned model (MLM$_{temp}$), the former requires additional training for the temporal attention mechanism, whereas SSCD require no additional training. This is particularly attractive from computational time and cost saving point-of-view. However, from Table 3 we see that there is a significant performance gap between the best unsupervised SCD method and the supervised XL-LEXEME. Although XL-LEXEME uses sense-labelled WiC data for fine-tuning sentence encoders, during inference only single point estimates of SCD scores are made. Therefore, it would be an interesting future research direction to explore the possibility of using SSCD at inference time with pre-trained XL-LEXEME, where multiple sets of sentences containing a target word is used to obtain a more reliable estimate of its SCD score.

---

[10]We also considered variants of SSCD that takes into account the number of sentences in $C_1$ and $C_2$ because the imbalance of the corpus size might affect context swapping process. Specifically, we normalise $N_1^w$ and $N_2^w$ respectively by $|\mathcal{C}_1|$ and $|\mathcal{C}_2|$ (i.e. the total numbers of sentences in each corpus), prior to computing $N_{swap}^w$ in Algorithm 1. However, in our preliminary investigations, we did not observe a significant improvement of performance due to this down sampling and believe it could be due to the fact that the datasets we used for evaluations are carefully sampled to have approximately equal numbers of sentences covering each point in time. Further results are shown in § D.2.

| Method | Spearman |
|---|---|
| Unsupervised | |
| Cosine (Rosin et al., 2022) | 0.467 |
| APD (Kutuzov and Giulianelli, 2020) | 0.479 |
| DSCD (Aida and Bollegala, 2023) | 0.529 |
| Temp. Att. (Rosin and Radinsky, 2022) | 0.548 |
| SSCD$_{rand}$, KL($C_1||C_2$) | 0.552 |
| SSCD$_{dist}$, DSCD | **0.563** |
| Supervised | |
| XL-LEXEME (Cassotti et al., 2023) | 0.757 |

Table 3: Comparison against strong baselines in SemEval-2020 Task 1 English. All methods except XL-LEXEME (Cassotti et al., 2023) start from the same model, MLM$_{temp}$ (Rosin et al., 2022).

### 4.3 Qualitative Analysis

We conduct an ablation study to further study the effect of context swapping and the swap rate. For this purpose, we use the following variants of our SSCD: 1) without context swapping (swap rate = 0.0), and 2) context swapping with extremely high or low swap rate (swap rate = 0.1 vs 1.0). Following Aida and Bollegala (2023), we select (a) words with the highest degree of semantic change (and labelled as semantically changed), and (b) words with the lowest degree of semantic change (and labelled as stable) by the annotators in the SemEval-2020 Task 1 for English SCD.

As shown in Table 4, we see that the use of context swapping improves the underestimation (*tip*) and overestimation (*fiction*) that occurs when context swapping is not used (i.e. swap rate = 0.0). Moreover, there is also a further improvement (*head*, *realtionship*, *fiction*) by using the optimum swap rate (swap rate = 0.4*).

However, we also see that SSCD cannot detect words which are rare (i.e. relatively low frequency of occurrence in the corpus) with novel or obsolete meanings (i.e. *bit*), or words used in different senses (*chairman* and *risk*). This is because SSCD assumes only one set of sibling embeddings per time period, which means that it can only roughly detect semantic changes in words. As mentioned in Aida and Bollegala (2023), we believe that separating the sets of sibling embeddings into sense levels (e.g. assuming mixed Gaussian distributions) will further improve the performance of SSCD. Although most of the target words in this benchmark are nouns and verbs, detecting words used in wider contexts, such as *chairman*, *risk* and adverbs, remains an interesting open problem for future work.

| Word | Gold | | Swap rate | | | |
|---|---|---|---|---|---|---|
| | rank | $\Delta$ | 0.0 | 0.1 | 0.4* | 1.0 |
| plane | 1 | ✓ | 1 | 1 | 1 | 1 |
| tip | 2 | ✓ | 25 | 7 | 2 | 2 |
| prop | 3 | ✓ | 3 | 2 | 4 | 20 |
| graft | 4 | ✓ | 2 | 9 | 12 | 36 |
| record | 5 | ✓ | 6 | 20 | 16 | 7 |
| stab | 7 | ✓ | 12 | 15 | 11 | 17 |
| bit | 9 | ✓ | 29 | 23 | 23 | 19 |
| head | 10 | ✓ | 33 | 12 | 13 | 32 |
| multitude | 30 | ✗ | 15 | 33 | 36 | 23 |
| savage | 31 | ✗ | 28 | 36 | 35 | 22 |
| contemplation | 32 | ✗ | 21 | 37 | 37 | 33 |
| tree | 33 | ✗ | 35 | 30 | 27 | 29 |
| relationship | 34 | ✗ | 17 | 35 | 34 | 21 |
| fiction | 35 | ✗ | 13 | 32 | 33 | 27 |
| chairman | 36 | ✗ | 4 | 26 | 14 | 3 |
| risk | 37 | ✗ | 7 | 16 | 22 | 14 |
| Spearman | 1.000 | | 0.130 | 0.596 | **0.627** | 0.164 |

Table 4: Ablation study on the words with the highest/lowest degree of semantic change labelled as changed/stable. $\Delta$ indicates the word is semantically changed (✓) or stable (✗). Swap rate $= 0.4^*$ is the optimal swap rate in this setting.

## 5 Conclusion

We proposed SSCD, a method that swaps contexts between two corpora for predicting semantic changes of words. Experimental results show that SSCD outperforms a previous method that uses context swapping for improving the reliability of SCD. Moreover, SSCD even with using only pretrained models, and not requiring fine-tuning achieves significant performance improvements compared to strong baselines for the unsupervised English SCD. We also discussed two perspectives on our SSCD: 1) We showed that modifying the context swapping method improves performance compared to random context swapping. 2) We proposed an unsupervised method to find the optimal context swapping-rate, and showed that it is relatively independent of the distance/divergence measure used in SSCD. As in future work, we will apply our method for time-dependent tasks such as temporal generalisation (Lazaridou et al., 2021) and temporal question answering (Dhingra et al., 2022). Furthermore, as our method and supervised methods are independent of each other, we are considering applying our method to the supervised SCD method.

## Limitations

In this paper, we show that our method can properly detect semantic changes in four languages (English, German, Swedish, and Latin) across two time spans (over 50 years and about five years). However, We do not conduct detecting *seasonal* semantic changes, which occur periodically and over extremely shorter time spans (e.g. few months). In e-commerce, keywords such as *scarf* used by users may refer to different products in different seasons. To detect and evaluate seasonal semantic changes, a training dataset and a list of words for evaluation need to be annotated in the future work.

## Ethics Statement

The goal of this paper is to detect semantic changes of words based on context swapping. For this purpose, we use publicly available SemEval 2020 Task 1 dataset, and do not collect or annotate any additional data by ourselves. Moreover, we are not aware of any social biases in the SemEval dataset that we use in our experiments. However, we also use pretrained MLMs in our experiments, which are known to encode unfair social biases such as racial or gender-related biases (Basta et al., 2019). Considering that the sets of sibling embeddings used in our proposed method are obtained from MLMs that may contain such social biases, the sensitivity of our method to such undesirable social biases needs to be further evaluated before it can be deployed in real-world applications used by human users.

## Acknowledgements

This work was supported by JST, the establishment of university fellowships towards the creation of science technology innovation, Grant Number JPMJFS2139. Danushka Bollegala holds concurrent appointments as a Professor at University of Liverpool and as an Amazon Scholar. This paper describes work performed at the University of Liverpool and is not associated with Amazon.

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

## A  Divergence Functions

We elaborate on two divergence functions. For simplicity, we denote two $d$-variate Gaussian distributions $\mathcal{N}(\boldsymbol{\mu}_1^w, \boldsymbol{\Sigma}_1^w)$ and $\mathcal{N}(\boldsymbol{\mu}_2^w, \boldsymbol{\Sigma}_2^w)$ as $\mathcal{N}_1^w$ and $\mathcal{N}_2^w$, respectively.

**Kullback-Leibler**

$$
\begin{aligned}
&\mathrm{KL}(\mathcal{N}_1^w || \mathcal{N}_2^w) \\
&= \frac{1}{2}\Big( \mathrm{tr}(\boldsymbol{\Sigma}_2^{w-1}\boldsymbol{\Sigma}_1^w) - d - \log \frac{\det(\boldsymbol{\Sigma}_1^w)}{\det(\boldsymbol{\Sigma}_2^w)} \\
&\quad + (\boldsymbol{\mu}_2^w - \boldsymbol{\mu}_1^w)^\top \boldsymbol{\Sigma}_2^{w-1}(\boldsymbol{\mu}_2^w - \boldsymbol{\mu}_1^w) \Big)
\end{aligned} \tag{1}
$$

**Jeffrey's**

$$\text{Jeff}(\mathcal{N}_1^w || \mathcal{N}_2^w)$$

$$= \frac{1}{2}\text{KL}(\mathcal{N}_1^w || \mathcal{N}_2^w) + \frac{1}{2}\text{KL}(\mathcal{N}_2^w || \mathcal{N}_1^w)$$

$$= \frac{1}{4}\Big( \text{tr}(\boldsymbol{\Sigma}_2^{w-1}\boldsymbol{\Sigma}_1^w) + \text{tr}(\boldsymbol{\Sigma}_1^{w-1}\boldsymbol{\Sigma}_2^w) - 2d \quad (2)$$

$$+ (\boldsymbol{\mu}_2^w - \boldsymbol{\mu}_1^w)^\top \boldsymbol{\Sigma}_2^{w-1}(\boldsymbol{\mu}_2^w - \boldsymbol{\mu}_1^w)$$

$$+ (\boldsymbol{\mu}_1^w - \boldsymbol{\mu}_2^w)^\top \boldsymbol{\Sigma}_1^{w-1}(\boldsymbol{\mu}_1^w - \boldsymbol{\mu}_2^w) \Big)$$

## B  Distance Functions

We elaborate on seven distance functions. In this part, $\boldsymbol{w}(i)$ denotes the $i$-dimensional value of the word vector $\boldsymbol{w}$ and $\overline{\boldsymbol{w}}$ denotes the vector subtracted by the average of all dimensional values.

**Bray-Curtis**

$$\psi(\boldsymbol{w}_1, \boldsymbol{w}_2) = \frac{\sum_{i \in d} |\boldsymbol{w}_1(i) - \boldsymbol{w}_2(i)|}{\sum_{i \in d} |\boldsymbol{w}_1(i) + \boldsymbol{w}_2(i)|} \quad (3)$$

**Canberra**

$$\psi(\boldsymbol{w}_1, \boldsymbol{w}_2) = \sum_{i \in d} \frac{|\boldsymbol{w}_1(i) - \boldsymbol{w}_2(i)|}{|\boldsymbol{w}_1(i)| + |\boldsymbol{w}_2(i)|} \quad (4)$$

**Chebyshev**

$$\psi(\boldsymbol{w}_1, \boldsymbol{w}_2) = \max_i |\boldsymbol{w}_1(i) - \boldsymbol{w}_2(i)| \quad (5)$$

**City Block**

$$\psi(\boldsymbol{w}_1, \boldsymbol{w}_2) = \sum_{i \in d} |\boldsymbol{w}_1(i) - \boldsymbol{w}_2(i)| \quad (6)$$

**Correlation**

$$\psi(\boldsymbol{w}_1, \boldsymbol{w}_2) = 1 - \frac{\overline{\boldsymbol{w}}_1 \cdot \overline{\boldsymbol{w}}_2}{||\overline{\boldsymbol{w}}_1||_2 \, ||\overline{\boldsymbol{w}}_2||_2} \quad (7)$$

**Cosine**

$$\psi(\boldsymbol{w}_1, \boldsymbol{w}_2) = 1 - \frac{\boldsymbol{w}_1 \cdot \boldsymbol{w}_2}{||\boldsymbol{w}_1||_2 \, ||\boldsymbol{w}_2||_2} \quad (8)$$

**Euclidean**

$$\psi(\boldsymbol{w}_1, \boldsymbol{w}_2) = ||\boldsymbol{w}_1 - \boldsymbol{w}_2||_2 \quad (9)$$

## C  Data Statistics

Data statistics are presented in Table 5. For the Liverpool FC benchmark, the statistics resulting from the pre-processing are shown. This data contains a variety of information, such as user names, ids, and timestamps, as well as the text. We extracted only the body text and used the NLTK library[11] to determine sentence boundaries and tokenise words.

[11] https://www.nltk.org/

## D  Full Results

### D.1  Effectiveness of Context Swapping

In § 4.1, we evaluate the performance of pretrained multilingual BERT with SSCD in the SemEval-2020 Task 1 and the Liverpool FC benchmarks. Tables 6-9 show the full results.

### D.2  Comparison against Strong Baselines

In § 4.2, we compare the performance of our SSCD against strong baselines in SemEval-2020 Task 1 English benchmark. Full results are shown in Table 10. In this setting, Table 11 shows the average and the standard deviation of the 20 seeds.

After that, we consider the context sampling strategies for our SSCD. Table 12 shows all the results of distance-based context swapping (SSCD$_{dist}$). Moreover, Tables 13 and 14 are the full results of considering the ratio of corpus size before context swapping methods SSCD$_{rand}$ and SSCD$_{dist}$.

| Dataset | Language | Time Period | #Targets | #Sentences | #Tokens | #Types |
|---|---|---|---|---|---|---|
| SemEval | English | 1810–1860 | 37 | 254k | 6.5M | 87k |
| | | 1960–2010 | | 354k | 6.7M | 150k |
| | German | 1800–1899 | 48 | 2.6M | 70.2M | 1.0M |
| | | 1946–1990 | | 3.5M | 72.3M | 2.3M |
| | Swedish | 1790–1830 | 30 | 3.4M | 71.0M | 1.9M |
| | | 1895–1903 | | 5.2M | 110.0M | 3.4M |
| Liverpool FC | English | 2011–2013 | 97 | 576k | 9.5M | 137k |
| | | 2017 | | 1.0M | 15.7M | 146k |

Table 5: Statistics of datasets.

| | Swap rate | | | | | | | | |
|---|---|---|---|---|---|---|---|---|---|
| Metric | 0.1 | 0.2 | 0.3 | 0.4 | 0.5 | 0.6 | 0.7 | 0.8 | 0.9 |
| Divergences | | | | | | | | | |
| $KL(C_1\|\|C_2)$ | **0.209** | 0.202 | 0.190 | 0.176 | 0.165 | 0.144 | 0.118 | 0.076 | 0.011 |
| $KL(C_2\|\|C_1)$ | **0.136** | 0.115 | 0.112 | 0.101 | 0.088 | 0.062 | 0.045 | 0.005 | 0.045 |
| $Jeff(C_1\|\|C_2)$ | **0.170** | 0.148 | 0.145 | 0.133 | 0.126 | 0.103 | 0.072 | 0.033 | 0.018 |
| Distance functions | | | | | | | | | |
| Bray-Curtis | 0.185 | **0.190** | 0.186 | 0.161 | 0.140 | 0.090 | 0.018 | 0.057 | 0.147 |
| Canberra | 0.223 | 0.245 | **0.246** | 0.217 | 0.189 | 0.117 | 0.010 | 0.091 | 0.156 |
| Chebyshev | 0.294 | 0.361 | 0.351 | **0.383** | 0.370 | 0.334 | 0.323 | 0.291 | 0.251 |
| City Block | **0.258** | 0.256 | 0.256 | 0.247 | 0.230 | 0.180 | 0.124 | 0.052 | 0.055 |
| Correlation | **0.183** | 0.171 | 0.162 | 0.161 | 0.155 | 0.131 | 0.104 | 0.061 | 0.002 |
| Cosine | **0.183** | 0.171 | 0.162 | 0.161 | 0.156 | 0.131 | 0.104 | 0.061 | 0.002 |
| Euclidean | 0.272 | 0.269 | **0.273** | 0.264 | 0.244 | 0.195 | 0.136 | 0.066 | 0.045 |
| DSCD (Aida and Bollegala, 2023) | | | | | | | | | |
| Bray-Curtis | 0.293 | 0.333 | 0.354 | 0.359 | **0.364** | 0.350 | 0.333 | 0.272 | 0.161 |
| Canberra | 0.183 | 0.177 | 0.224 | 0.246 | **0.282** | 0.258 | 0.262 | 0.199 | 0.103 |
| Chebyshev | 0.080 | 0.120 | 0.178 | 0.180 | **0.196** | **0.196** | 0.177 | 0.104 | 0.101 |
| City Block | 0.212 | 0.261 | 0.285 | **0.312** | 0.301 | 0.295 | 0.284 | 0.264 | 0.211 |
| Correlation | 0.154 | 0.234 | 0.266 | 0.277 | **0.291** | 0.287 | 0.268 | 0.233 | 0.173 |
| Cosine | 0.229 | 0.254 | 0.319 | 0.331 | **0.336** | 0.305 | 0.283 | 0.258 | 0.175 |
| Euclidean | 0.293 | 0.313 | 0.340 | **0.356** | 0.334 | 0.327 | 0.310 | 0.283 | 0.216 |

Table 6: Results within MLM$_{pre}$ with SSCD$_{rand}$ in SemEval-2020 Task 1 English. All values are averages over 20 seeds.

| Metric | Swap rate | | | | | | | | |
|---|---|---|---|---|---|---|---|---|---|
| | 0.1 | 0.2 | 0.3 | 0.4 | 0.5 | 0.6 | 0.7 | 0.8 | 0.9 |
| Divergences | | | | | | | | | |
| $\mathrm{KL}(C_1||C_2)$ | 0.511 | 0.536 | **0.547** | 0.536 | 0.537 | 0.539 | 0.533 | 0.520 | 0.490 |
| $\mathrm{KL}(C_2||C_1)$ | 0.510 | 0.526 | **0.542** | 0.539 | 0.537 | 0.535 | 0.526 | 0.509 | 0.477 |
| $\mathrm{Jeff}(C_1||C_2)$ | 0.515 | 0.533 | **0.547** | 0.540 | 0.540 | 0.538 | 0.531 | 0.517 | 0.484 |
| Distance functions | | | | | | | | | |
| Bray-Curtis | 0.486 | 0.522 | 0.560 | 0.576 | 0.580 | **0.597** | 0.583 | 0.529 | 0.434 |
| Canberra | 0.370 | 0.431 | 0.474 | 0.499 | 0.512 | **0.550** | 0.534 | 0.459 | 0.332 |
| Chebyshev | 0.236 | 0.288 | 0.355 | 0.385 | 0.398 | **0.447** | 0.462 | 0.462 | 0.439 |
| City Block | 0.337 | 0.358 | 0.381 | 0.386 | 0.388 | 0.414 | 0.417 | **0.428** | 0.369 |
| Correlation | 0.563 | 0.577 | 0.591 | **0.592** | 0.586 | 0.586 | 0.577 | 0.554 | 0.525 |
| Cosine | 0.563 | 0.577 | 0.591 | **0.592** | 0.586 | 0.586 | 0.577 | 0.554 | 0.525 |
| Euclidean | 0.348 | 0.364 | 0.392 | 0.394 | 0.397 | 0.423 | 0.422 | **0.436** | 0.377 |
| DSCD (Aida and Bollegala, 2023) | | | | | | | | | |
| Bray-Curtis | 0.388 | 0.432 | 0.473 | **0.476** | 0.471 | 0.473 | 0.466 | 0.441 | 0.407 |
| Canberra | 0.284 | 0.360 | 0.420 | 0.429 | 0.424 | **0.435** | 0.429 | 0.403 | 0.359 |
| Chebyshev | 0.006 | 0.062 | 0.126 | 0.172 | 0.197 | **0.204** | 0.201 | 0.180 | 0.158 |
| City Block | 0.338 | 0.348 | 0.381 | 0.396 | 0.381 | 0.390 | **0.400** | 0.389 | 0.366 |
| Correlation | 0.373 | 0.427 | 0.457 | **0.460** | 0.443 | 0.437 | 0.435 | 0.408 | 0.375 |
| Cosine | 0.262 | 0.338 | 0.400 | **0.420** | 0.413 | 0.406 | 0.406 | 0.378 | 0.344 |
| Euclidean | 0.274 | 0.315 | 0.357 | 0.383 | 0.369 | 0.378 | **0.389** | 0.383 | 0.356 |

Table 7: Results within $\mathrm{MLM}_{pre}$ with $\mathrm{SSCD}_{rand}$ in SemEval-2020 Task 1 German. All values are averages over 20 seeds.

| Metric | Swap rate | | | | | | | | |
|---|---|---|---|---|---|---|---|---|---|
| | 0.1 | 0.2 | 0.3 | 0.4 | 0.5 | 0.6 | 0.7 | 0.8 | 0.9 |
| Divergences | | | | | | | | | |
| $KL(C_1||C_2)$ | 0.104 | 0.083 | 0.072 | 0.058 | 0.038 | 0.019 | 0.004 | 0.029 | **0.127** |
| $KL(C_2||C_1)$ | 0.110 | 0.083 | 0.088 | 0.081 | 0.063 | 0.037 | 0.035 | 0.014 | **0.114** |
| $Jeff(C_1||C_2)$ | 0.112 | 0.092 | 0.087 | 0.079 | 0.060 | 0.030 | 0.014 | 0.025 | **0.125** |
| Distance functions | | | | | | | | | |
| Bray-Curtis | 0.002 | 0.020 | 0.014 | 0.015 | **0.022** | **0.022** | 0.020 | 0.019 | 0.006 |
| Canberra | 0.006 | 0.004 | 0.003 | 0.008 | 0.016 | 0.047 | 0.069 | **0.109** | 0.105 |
| Chebyshev | 0.107 | 0.094 | 0.071 | 0.084 | 0.087 | 0.106 | 0.109 | **0.142** | 0.116 |
| City Block | 0.154 | 0.152 | 0.146 | 0.155 | 0.167 | 0.180 | 0.229 | 0.227 | **0.230** |
| Correlation | 0.036 | 0.042 | 0.054 | 0.061 | 0.064 | 0.095 | **0.101** | 0.068 | 0.036 |
| Cosine | 0.036 | 0.042 | 0.054 | 0.061 | 0.064 | 0.095 | **0.101** | 0.068 | 0.036 |
| Euclidean | 0.151 | 0.146 | 0.137 | 0.145 | 0.163 | 0.173 | 0.229 | 0.232 | **0.234** |
| DSCD (Aida and Bollegala, 2023) | | | | | | | | | |
| Bray-Curtis | **0.068** | 0.015 | 0.015 | 0.018 | 0.037 | 0.050 | 0.059 | 0.057 | 0.054 |
| Canberra | 0.013 | 0.011 | 0.018 | 0.048 | 0.067 | 0.082 | 0.105 | **0.119** | 0.095 |
| Chebyshev | 0.162 | 0.137 | 0.143 | 0.176 | 0.197 | **0.199** | 0.171 | 0.140 | 0.160 |
| City Block | 0.094 | 0.121 | 0.115 | 0.117 | 0.113 | 0.111 | 0.116 | **0.123** | 0.119 |
| Correlation | **0.127** | 0.112 | 0.103 | 0.102 | 0.108 | 0.097 | 0.077 | 0.081 | 0.093 |
| Cosine | **0.090** | 0.072 | 0.075 | 0.068 | 0.088 | 0.076 | 0.050 | 0.061 | 0.066 |
| Euclidean | **0.109** | 0.093 | 0.106 | 0.093 | 0.090 | 0.092 | 0.084 | 0.089 | 0.060 |

Table 8: Results within $MLM_{pre}$ with $SSCD_{rand}$ in SemEval-2020 Task 1 Swedish. All values are averages over 20 seeds.

| Metric | Swap rate | | | | | | | | |
|---|---|---|---|---|---|---|---|---|---|
| | 0.1 | 0.2 | 0.3 | 0.4 | 0.5 | 0.6 | 0.7 | 0.8 | 0.9 |
| Divergences | | | | | | | | | |
| $KL(C_1||C_2)$ | 0.052 | 0.198 | 0.268 | 0.291 | 0.305 | 0.460 | 0.464 | **0.470** | 0.462 |
| $KL(C_2||C_1)$ | 0.047 | 0.157 | 0.235 | 0.247 | 0.259 | **0.396** | 0.394 | 0.394 | 0.393 |
| $Jeff(C_1||C_2)$ | 0.051 | 0.172 | 0.260 | 0.277 | 0.291 | 0.438 | 0.438 | 0.439 | **0.440** |
| Distance functions | | | | | | | | | |
| Bray-Curtis | 0.027 | 0.186 | 0.261 | 0.278 | 0.293 | 0.441 | 0.448 | 0.462 | **0.476** |
| Canberra | 0.018 | 0.135 | 0.202 | 0.228 | 0.245 | 0.375 | 0.391 | 0.405 | **0.436** |
| Chebyshev | 0.025 | 0.085 | 0.130 | 0.166 | 0.177 | 0.334 | 0.343 | 0.362 | **0.384** |
| City Block | 0.000 | 0.161 | 0.248 | 0.273 | 0.294 | 0.465 | 0.467 | 0.473 | **0.489** |
| Correlation | 0.065 | 0.217 | 0.298 | 0.309 | 0.317 | 0.475 | 0.477 | 0.480 | **0.488** |
| Cosine | 0.065 | 0.217 | 0.298 | 0.309 | 0.317 | 0.475 | 0.477 | 0.480 | **0.488** |
| Euclidean | 0.001 | 0.161 | 0.249 | 0.275 | 0.294 | 0.466 | 0.469 | 0.470 | **0.492** |
| DSCD (Aida and Bollegala, 2023) | | | | | | | | | |
| Bray-Curtis | 0.116 | 0.173 | 0.201 | 0.231 | 0.213 | 0.342 | 0.344 | 0.352 | **0.354** |
| Canberra | 0.098 | 0.149 | 0.208 | 0.227 | 0.215 | 0.328 | 0.338 | 0.350 | **0.358** |
| Chebyshev | 0.150 | 0.163 | 0.189 | 0.210 | 0.221 | 0.279 | 0.264 | 0.268 | **0.291** |
| City Block | 0.174 | 0.181 | 0.210 | 0.229 | 0.226 | **0.364** | 0.358 | 0.353 | 0.357 |
| Correlation | 0.087 | 0.158 | 0.177 | 0.212 | 0.209 | **0.358** | 0.345 | 0.340 | 0.349 |
| Cosine | 0.093 | 0.157 | 0.185 | 0.214 | 0.207 | **0.354** | 0.342 | 0.340 | 0.344 |
| Euclidean | 0.170 | 0.199 | 0.218 | 0.236 | 0.233 | **0.362** | 0.356 | 0.350 | 0.353 |

Table 9: Results within $MLM_{pre}$ with $SSCD_{rand}$ in Liverpool FC. All values are averages of 20 seeds.

| Metric | Swap rate | | | | | | | | |
|---|---|---|---|---|---|---|---|---|---|
| | 0.1 | 0.2 | 0.3 | 0.4 | 0.5 | 0.6 | 0.7 | 0.8 | 0.9 |
| | | | | Divergences | | | | | |
| $KL(C_1\|\|C_2)$ | 0.461 | 0.536 | 0.549 | **0.552** | 0.543 | 0.526 | 0.483 | 0.396 | 0.239 |
| $KL(C_2\|\|C_1)$ | 0.433 | 0.503 | 0.509 | **0.516** | 0.514 | 0.497 | 0.439 | 0.369 | 0.234 |
| $Jeff(C_1\|\|C_2)$ | 0.450 | 0.522 | 0.528 | **0.534** | 0.529 | 0.516 | 0.460 | 0.383 | 0.239 |
| | | | | Distance functions | | | | | |
| Bray-Curtis | 0.267 | 0.392 | 0.395 | **0.423** | 0.391 | 0.335 | 0.259 | 0.140 | 0.020 |
| Canberra | 0.194 | 0.283 | 0.296 | **0.345** | 0.292 | 0.229 | 0.148 | 0.063 | 0.073 |
| Chebyshev | 0.158 | 0.273 | 0.317 | **0.372** | 0.341 | 0.351 | 0.276 | 0.197 | 0.081 |
| City Block | 0.282 | 0.413 | 0.414 | **0.443** | 0.407 | 0.349 | 0.273 | 0.148 | 0.007 |
| Correlation | 0.355 | 0.441 | 0.445 | **0.471** | **0.471** | 0.466 | 0.437 | 0.350 | 0.153 |
| Cosine | 0.355 | 0.441 | 0.445 | **0.471** | **0.471** | 0.466 | 0.437 | 0.350 | 0.153 |
| Euclidean | 0.296 | 0.419 | 0.419 | **0.453** | 0.416 | 0.358 | 0.272 | 0.148 | 0.004 |
| | | | | DSCD (Aida and Bollegala, 2023) | | | | | |
| Bray-Curtis | 0.149 | 0.331 | 0.356 | **0.367** | 0.336 | 0.293 | 0.214 | 0.134 | 0.057 |
| Canberra | 0.233 | 0.341 | 0.385 | **0.378** | 0.354 | 0.318 | 0.267 | 0.202 | 0.111 |
| Chebyshev | 0.029 | 0.026 | 0.030 | **0.120** | 0.112 | 0.118 | 0.118 | 0.122 | 0.146 |
| City Block | 0.245 | 0.336 | **0.411** | 0.400 | 0.377 | 0.335 | 0.250 | 0.195 | 0.120 |
| Correlation | 0.185 | 0.315 | 0.352 | **0.365** | 0.339 | 0.301 | 0.224 | 0.160 | 0.089 |
| Cosine | 0.212 | 0.332 | **0.383** | 0.375 | 0.356 | 0.312 | 0.253 | 0.212 | 0.125 |
| Euclidean | 0.088 | 0.237 | **0.315** | 0.313 | 0.298 | 0.253 | 0.210 | 0.154 | 0.074 |

Table 10: Results within MLM$_{temp}$ with SSCD$_{rand}$ in SemEval-2020 Task 1 English. All values are averages over 20 seeds.

| Swap rate | Divergences | | |
|---|---|---|---|
| | $KL(C_1\|\|C_2)$ | $KL(C_2\|\|C_1)$ | $Jeff(C_1\|\|C_2)$ |
| 0.1 | 0.461±0.080 | 0.433±0.080 | 0.450±0.076 |
| 0.2 | 0.536±0.074 | 0.503±0.071 | 0.522±0.077 |
| 0.3 | 0.549±0.064 | 0.509±0.065 | 0.528±0.063 |
| 0.4 | **0.552**±0.037 | **0.516**±0.041 | **0.534**±0.037 |
| 0.5 | 0.543±0.037 | 0.514±0.032 | 0.529±0.032 |
| 0.6 | 0.526±0.021 | 0.497±0.026 | 0.516±0.024 |
| 0.7 | 0.483±0.027 | 0.439±0.030 | 0.460±0.030 |
| 0.8 | 0.396±0.044 | 0.369±0.049 | 0.383±0.047 |
| 0.9 | 0.239±0.035 | 0.234±0.043 | 0.239±0.037 |

Table 11: Full results of divergence functions within MLM$_{temp}$ with SSCD$_{rand}$ in SemEval-2020 Task 1 English.

| Metric | Swap rate | | | | | | | | |
|---|---|---|---|---|---|---|---|---|---|
| | 0.1 | 0.2 | 0.3 | 0.4 | 0.5 | 0.6 | 0.7 | 0.8 | 0.9 |
| Divergences | | | | | | | | | |
| $KL(C_1||C_2)$ | 0.501 | **0.507** | 0.490 | 0.325 | 0.124 | 0.022 | 0.069 | 0.147 | 0.214 |
| $KL(C_2||C_1)$ | 0.463 | **0.466** | 0.427 | 0.235 | 0.014 | 0.144 | 0.218 | 0.039 | 0.048 |
| $Jeff(C_1||C_2)$ | 0.475 | **0.496** | 0.474 | 0.291 | 0.051 | 0.101 | 0.079 | 0.003 | 0.117 |
| Distance functions | | | | | | | | | |
| Bray-Curtis | 0.378 | **0.396** | 0.355 | 0.179 | 0.004 | 0.241 | 0.229 | 0.166 | 0.046 |
| Canberra | 0.304 | **0.420** | 0.249 | 0.172 | 0.015 | 0.261 | 0.306 | 0.223 | 0.086 |
| Chebyshev | 0.268 | **0.492** | 0.307 | 0.088 | 0.121 | 0.269 | 0.315 | 0.336 | 0.243 |
| City Block | 0.392 | **0.395** | 0.363 | 0.201 | 0.003 | 0.241 | 0.232 | 0.178 | 0.032 |
| Correlation | **0.454** | 0.434 | 0.430 | 0.226 | 0.010 | 0.228 | 0.112 | 0.088 | 0.037 |
| Cosine | **0.454** | 0.434 | 0.430 | 0.226 | 0.010 | 0.228 | 0.112 | 0.088 | 0.037 |
| Euclidean | **0.401** | 0.398 | 0.356 | 0.162 | 0.020 | 0.251 | 0.145 | 0.125 | 0.073 |
| DSCD (Aida and Bollegala, 2023) | | | | | | | | | |
| Bray-Curtis | 0.242 | 0.298 | 0.174 | 0.004 | 0.028 | 0.253 | 0.394 | 0.444 | **0.474** |
| Canberra | 0.349 | **0.557** | 0.253 | 0.163 | 0.084 | 0.328 | 0.226 | 0.395 | 0.426 |
| Chebyshev | 0.266 | 0.106 | 0.003 | 0.110 | 0.073 | 0.054 | 0.114 | **0.267** | 0.263 |
| City Block | 0.259 | 0.264 | 0.210 | 0.099 | 0.030 | 0.268 | 0.303 | **0.414** | 0.372 |
| Correlation | 0.334 | 0.427 | 0.220 | 0.054 | 0.051 | 0.231 | 0.362 | **0.563** | 0.492 |
| Cosine | **0.528** | 0.490 | 0.219 | 0.011 | 0.102 | 0.295 | 0.342 | 0.499 | 0.489 |
| Euclidean | 0.402 | 0.310 | 0.233 | 0.051 | 0.073 | 0.111 | 0.312 | 0.447 | **0.467** |

Table 12: Results within $MLM_{temp}$ with $SSCD_{dist}$ in SemEval-2020 Task 1 English. All values are averages over 20 seeds.

| Metric | Swap rate | | | | | | | | |
|---|---|---|---|---|---|---|---|---|---|
| | 0.1 | 0.2 | 0.3 | 0.4 | 0.5 | 0.6 | 0.7 | 0.8 | 0.9 |
| Divergences | | | | | | | | | |
| $KL(C_1||C_2)$ | 0.445 | 0.493 | 0.507 | 0.514 | 0.513 | **0.523** | 0.518 | 0.512 | 0.488 |
| $KL(C_2||C_1)$ | 0.411 | 0.467 | 0.475 | 0.488 | 0.486 | **0.492** | 0.485 | 0.472 | 0.460 |
| $Jeff(C_1||C_2)$ | 0.428 | 0.484 | 0.490 | 0.505 | 0.501 | **0.512** | 0.504 | 0.490 | 0.470 |
| Distance functions | | | | | | | | | |
| Bray-Curtis | 0.282 | 0.376 | 0.370 | 0.400 | 0.407 | **0.409** | 0.351 | 0.293 | 0.195 |
| Canberra | 0.238 | 0.287 | 0.271 | 0.330 | **0.334** | 0.321 | 0.239 | 0.191 | 0.083 |
| Chebyshev | 0.208 | 0.290 | 0.334 | 0.403 | 0.387 | **0.412** | 0.366 | 0.304 | 0.261 |
| City Block | 0.299 | 0.392 | 0.382 | 0.411 | 0.422 | **0.423** | 0.366 | 0.303 | 0.204 |
| Correlation | 0.365 | 0.434 | 0.444 | 0.453 | 0.467 | 0.469 | **0.472** | 0.452 | 0.396 |
| Cosine | 0.365 | 0.434 | 0.444 | 0.453 | 0.467 | 0.469 | **0.472** | 0.452 | 0.396 |
| Euclidean | 0.309 | 0.400 | 0.391 | 0.421 | **0.429** | 0.426 | 0.371 | 0.310 | 0.214 |
| DSCD (Aida and Bollegala, 2023) | | | | | | | | | |
| Bray-Curtis | 0.375 | **0.477** | 0.457 | 0.471 | 0.437 | 0.436 | 0.410 | 0.366 | 0.351 |
| Canberra | 0.007 | 0.209 | 0.258 | 0.293 | **0.324** | 0.293 | 0.270 | 0.242 | 0.181 |
| Chebyshev | **0.158** | 0.054 | 0.005 | 0.114 | 0.119 | 0.101 | 0.150 | 0.146 | 0.146 |
| City Block | 0.205 | 0.336 | 0.365 | **0.403** | 0.371 | 0.374 | 0.327 | 0.316 | 0.279 |
| Correlation | 0.263 | 0.396 | 0.401 | **0.440** | 0.411 | 0.381 | 0.356 | 0.329 | 0.298 |
| Cosine | 0.309 | 0.438 | 0.436 | **0.448** | 0.431 | 0.412 | 0.379 | 0.355 | 0.329 |
| Euclidean | 0.193 | 0.351 | 0.366 | **0.387** | 0.385 | 0.377 | 0.347 | 0.337 | 0.305 |

Table 13: Results within MLM$_{temp}$ with normalized SSCD$_{rand}$ in SemEval-2020 Task 1 English. All values are averages over 20 seeds.

| Metric | Swap rate | | | | | | | | |
|---|---|---|---|---|---|---|---|---|---|
| | 0.1 | 0.2 | 0.3 | 0.4 | 0.5 | 0.6 | 0.7 | 0.8 | 0.9 |
| Divergences | | | | | | | | | |
| $KL(C_1||C_2)$ | 0.442 | 0.482 | **0.538** | 0.507 | 0.420 | 0.223 | 0.151 | 0.079 | 0.237 |
| $KL(C_2||C_1)$ | 0.415 | 0.449 | **0.485** | 0.420 | 0.232 | 0.060 | 0.048 | 0.037 | 0.019 |
| $Jeff(C_1||C_2)$ | 0.434 | 0.473 | **0.516** | 0.500 | 0.345 | 0.128 | 0.034 | 0.041 | 0.145 |
| Distance functions | | | | | | | | | |
| Bray-Curtis | 0.326 | **0.408** | 0.403 | 0.340 | 0.246 | 0.053 | 0.094 | 0.113 | 0.107 |
| Canberra | 0.259 | **0.405** | 0.313 | 0.246 | 0.153 | 0.036 | 0.167 | 0.212 | 0.141 |
| Chebyshev | 0.289 | 0.436 | **0.499** | 0.255 | 0.009 | 0.242 | 0.287 | 0.312 | 0.237 |
| City Block | 0.336 | **0.413** | 0.409 | 0.338 | 0.243 | 0.046 | 0.104 | 0.112 | 0.105 |
| Correlation | 0.385 | 0.424 | **0.461** | 0.428 | 0.282 | 0.049 | 0.018 | 0.021 | 0.083 |
| Cosine | 0.385 | 0.424 | **0.461** | 0.428 | 0.282 | 0.049 | 0.018 | 0.021 | 0.083 |
| Euclidean | 0.346 | 0.405 | **0.413** | 0.349 | 0.232 | 0.031 | 0.076 | 0.049 | 0.163 |
| DSCD (Aida and Bollegala, 2023) | | | | | | | | | |
| Bray-Curtis | **0.475** | 0.377 | 0.419 | 0.217 | 0.057 | 0.159 | 0.254 | 0.370 | 0.470 |
| Canberra | 0.305 | **0.449** | 0.271 | 0.217 | 0.097 | 0.042 | 0.209 | 0.375 | 0.313 |
| Chebyshev | 0.190 | 0.105 | 0.042 | 0.022 | 0.211 | 0.259 | 0.157 | **0.402** | 0.251 |
| City Block | 0.336 | 0.394 | 0.304 | 0.265 | 0.089 | 0.006 | 0.186 | 0.441 | **0.463** |
| Correlation | 0.414 | 0.440 | 0.344 | 0.186 | 0.120 | 0.158 | 0.287 | **0.480** | 0.443 |
| Cosine | 0.348 | 0.346 | 0.244 | 0.114 | 0.050 | 0.176 | 0.236 | 0.369 | **0.424** |
| Euclidean | 0.302 | 0.430 | 0.244 | 0.089 | 0.016 | 0.277 | 0.377 | 0.327 | **0.451** |

Table 14: Results within $MLM_{temp}$ with normalized $SSCD_{dist}$ in SemEval-2020 Task 1 English. All values are averages over 20 seeds.