# OpenReview forum: "$\textit{Swap and Predict}$ -- Predicting the Semantic Changes in Words across Corpora by Context Swapping"
_EMNLP/2023/Conference — EMNLP 2023 Findings_

### Official Review · Reviewer_SSFX · 2023-08-07

**Soundness:** 3

**Excitement:**

2: Mediocre: This paper makes marginal contributions (vs non-contemporaneous work), so I would rather not see it in the conference.

**Paper Topic And Main Contributions:**

This paper proposes a method (SSCD) to measure semantic shift for individual words between two given corpora. It relies on the distance between sibling distributions (Aida and Bollegala, 2023) to compare words and on a context-swapping test similar to the permutation significance test used by Liu et al. (2021). The experiments show that SSCD sometimes correlates on par or better than previous approaches with human ranking of words by their degrees of semantic shifts on two datasets.

**Questions For The Authors:**

* The statement on line 113 is not precise. The authors should specify which chapter and section of Dekking et al. (2005) they're referring to.

* In the experiments where the swapping rate is optimized (Fig 2), why doesn't the divergence reach 0? Is this related to random noise and differences in size corpora and other factors for which the swapping rate compensates? How is the swapping rate related to permutations of combined corpora as is standard in permutation significance tests?

* I don't understand the experiment with swap-rate = 0. Do you still use Algorithm 2 for Table 4 or the distance $e_{original}$ only?


**Reasons To Accept:**

* The idea is simple and somewhat novel up to my knowledge.
* It performs well on two datasets.

**Reasons To Reject:**

* The motivation given between lines 104 and 125 needs some clarification. The authors write that "previous approach provides only a single estimate" based on corpora C1 and C2. But this is the only option if C1 and C2 are seen as samples from some distribution. The sampling distribution cannot be used unless multiple samples can be drawn from the distribution.

* The proposed SSCD seems more like a test for the difference between two samples similar to the permutation test for statistical significance used by Liu et al. (2021) but without relying on a predefined p-value. While the authors point to these similarities and discuss the limitations to overcome starting from line 214, it is not clear how this is achieved by SSCD. The authors only point to increased performance on SemEval but no insight is given.

* The authors discuss rejecting (or not) the null hypothesis in section 3.1 based on the value of $|e_{original} - e_{swap}|$ but no further details are given. This score is interpreted as a degree of semantic shift and used only to rank words.

* While Liu et al. (2021) fine-tune the LM before computing word embeddings, fine-tuning can be skipped. On the other hand, the authors can also test SSCD with fine-tuned LM. Therefore, the argument about efficiency made in section 4.1 is not valid unless some principled reason is provided for the choice of fine-tuning or lack thereof. Liu et al. (2021) and Martinc et al. (2020) also don't discuss this question.

* The superiority of the proposed approach in (some of) the tasks described in the introduction is yet to be demonstrated.

* I understand that distributional models are models of word meaning, see, for instance, Sahlgren (2008) for a discussion. However, in my opinion, detecting new meanings of words using paradigmatic models is not a completely automated process. It often requires human validation and domain knowledge. There are instances where shifts in distributional properties of words are not necessarily related to new meanings but rather reflect other factors or linguistic phenomena. These factors include trends and use in pop culture, regional variations, slang, neologisms, acronyms, lexical borrowing or simply noise in the used corpora. I think the paper can benefit from a discussion (and analysis) of this issue.


Reference:
The distributional hypothesis, Magnus Sahlgren, Italian Journal of Linguistics, 2008


**Reproducibility:**

4: Could mostly reproduce the results, but there may be some variation because of sample variance or minor variations in their interpretation of the protocol or method.

**Reviewer Confidence:**

3: Pretty sure, but there's a chance I missed something. Although I have a good feel for this area in general, I did not carefully check the paper's details, e.g., the math, experimental design, or novelty.

---

> ### Author Rebuttal · Authors · 2023-08-27
>
> We appreciate your time and effort to provide insightful feedback on our paper. Below, we provide a summary of the main points and our corresponding responses.
>
> Q1: *The sampling distribution cannot be used unless multiple samples can be drawn from the distribution.*
> A1: Even though only two snapshots C1 and C2 might be given, we can still obtain subsamples (with replacement) to make a more reliable estimate, as done by our proposed method SSCD. Therefore, we disagree with the statement "this is the only option if C1 and C2 are seen as samples from some distribution".
>
> Q2: *from line 214, it is not clear how this is achieved by SSCD*
> A2: This is already described in Lines 311-322 in the paper. We can perform a bootstrapping test using the value $|e_{original} - e_{swap}|$ if we would like to compute p-values. However, note that this is not necessary for the graded semantic change detection tasks we conduct in the paper.
>
> Q3: *This score is interpreted as a degree of semantic shift and used only to rank words.*
> A3: Please see the response to the previous (Q2) comment/question.
>
> Q4: *the argument about efficiency made in section 4.1 is not valid*
> A4: Fine-tuning of large MLMs is indeed very expensive. Parameter-efficient fine-tuning methods such as LoRA [Hu et al., ICLR 2022] have been proposed specifically due to this efficiency reason. Therefore, we believe it is an important distinction especially when using semantic change detection (SCD) methods at scale using large MLMs.
>
> Q5: *The superiority of the proposed approach in (some of) the tasks described in the introduction is yet to be demonstrated.*
> A5: It is not clear what are these "some of tasks" you are referring to. We have conducted evaluations on two different datasets and on three different languages, which is over and above the evaluations conducted in prior work on this topic.
>
> Q6: *However, in my opinion, detecting new meanings of words using paradigmatic models is not a completely automated process.*
> A6: We do not claim SCD is a fully automatic process either. As you point out, although distributional changes are usually associated with semantic changes of words, there could be other reasons as well. On the other hand, the SSCD scores that we compute will help humans (linguists for example) to filter candidate words that might have undergone a semantic change for their manual investigation, which will save their time and effort.
>
> Q7: *The statement on line 113 is not precise. The authors should specify which chapter and section of Dekking et al. (2005) they're referring to.*
> A7: We refer to section 17.2 in Dekking et al. (2005). We will include this information in the camera ready version as well, if accepted.
>
> Q8: *In the experiments where the swapping rate is optimized (Fig 2), why doesn't the divergence reach 0?*
> A8: Yes, this is exactly because of the random noise in the data.
>
> Q9: *I don't understand the experiment with swap-rate = 0. Do you still use Algorithm 2 for Table 4 or the distance $e_{original}$ only?*
> A9: Yes, we use $e_{original}$ only when swap rate = 0.

---

### Official Review · Reviewer_Wzo4 · 2023-08-09

**Typos Grammar Style And Presentation Improvements:** 518-519
**Soundness:** 4

**Excitement:**

3: Ambivalent: It has merits (e.g., it reports state-of-the-art results, the idea is nice), but there are key weaknesses (e.g., it describes incremental work), and it can significantly benefit from another round of revision. However, I won't object to accepting it if my co-reviewers champion it.

**Paper Topic And Main Contributions:**

This paper tackles the task of semantic change detection (by way of the standard SemEval 2020 Task 1). It introduces a new method that involves obtaining contextual embeddings for the target word in corpora from different times, and comparing their distributions. Various different distance and divergence measures are tried to compare the distributions. The method beat the state-of-the-art result for this task, and the authors compare the metrics to various other recent approaches to this task. The task is done for English, German and Swedish, and an advantage of the method is that there is no need to tune a model.

**Reasons To Accept:**

The paper contributes a methodological innovation in the area of semantic change detection and beats the state of the art at the time of this paper's submission. It is demonstrated for several languages, and appears to work well also for low-frequency words which is important in specific domains. Classification decisions appear to be statisticlly significant.


**Reasons To Reject:**

While the performance looks good, the method seems to depend on contextual embeddings derived from models pretrained on contemporary language data. It seems to me that these contextual embeddings should be less accurate, the further away we get from the training data. It might thus be the case that this method would not work well if we are not dealing with a change from the relatively recent past from the present. How would the approach perform on semantic changes between 1800-1850 rather than 1970-2020? Probably a lot worse, but this is not really explored. Of course, there is no gold standard for this, but the limitation could have been discussed or perhaps qualitatively explored. There is also little intrinsic exploration of the correctness of the meaning representations of these contextual embeddings in the paper.

Although the state-of-the-art is beaten and there is methodological innovation, this methodological innovation is limited in scope, relying on some other recent work. The swapping had been done before, just not for low-frequency words and without model tuning.

As the authors note, the method is only applied to change detection over relatively long periods of time, in which the task is easier than e.g. recent temporal generalization

**Reproducibility:**

4: Could mostly reproduce the results, but there may be some variation because of sample variance or minor variations in their interpretation of the protocol or method.

**Reviewer Confidence:**

3: Pretty sure, but there's a chance I missed something. Although I have a good feel for this area in general, I did not carefully check the paper's details, e.g., the math, experimental design, or novelty.

---

> ### Author Rebuttal · Authors · 2023-08-27
>
> We appreciate your time and effort to provide insightful feedback on our paper. Below, we provide a summary of the main points and our corresponding responses.
>
> Q1: *The method would not work for changes that happened in the past.*
> A1: Although the model performance depends on the MLM being used, this is orthogonal to the main technical contributions we make in this paper, which is a swapping-based sampling method for semantic change detection (SCD). If we know the duration within which we must detect semantic changes of words, we can pre-train/fine-tune MLMs on historical data from that time period to obtain an MLM that is better suited for the SCD task.
>
> Q2: *There is also little intrinsic exploration of the correctness of the meaning representations of these contextual embeddings in the paper.*
> A2: According to the distributional hypothesis, we assume that contextual embeddings to reflect word meanings. If there are more accurate models for word meaning representation, we can use those instead of contextualised word embeddings to further improve the accuracy of SCD with our proposed method.
>
> Q3: *As the authors note, the method is only applied to change detection over relatively long periods of time, in which the task is easier than e.g. recent temporal generalization.*
> A3: Please note that we not only conduct long-term SCD such as over 50 years, but also short-term SCD of around 5 years, as in the Liverpool FC benchmark.

---

### Official Review · Reviewer_UWZX · 2023-08-10

**Soundness:** 3

**Excitement:**

2: Mediocre: This paper makes marginal contributions (vs non-contemporaneous work), so I would rather not see it in the conference.

**Missing References:**

- [1] Guy D. Rosin and Kira Radinsky. 2022. Temporal Attention for Language Models. In Findings of the Association for Computational Linguistics: NAACL 2022, pages 1498–1508, Seattle, United States. Association for Computational Linguistics.
- [2] Andrey Kutuzov, Erik Velldal, and Lilja Øvrelid. 2022. Contextualized embeddings for semantic change detection: Lessons learned. In Northern European Journal of Language Technology, Volume 8, Copenhagen, Denmark. Northern European Association of Language Technology.
- [3] Andrey Kutuzov and Mario Giulianelli. 2020. UiO-UvA at SemEval-2020 Task 1: Contextualised Embeddings for Lexical Semantic Change Detection. In Proceedings of the Fourteenth Workshop on Semantic Evaluation, pages 126–134, Barcelona (online). International Committee for Computational Linguistics.
- [4] Pierluigi Cassotti, Lucia Siciliani, Marco DeGemmis, Giovanni Semeraro, and Pierpaolo Basile. 2023. XL-LEXEME: WiC Pretrained Model for Cross-Lingual LEXical sEMantic changE. In Proceedings of the 61st Annual Meeting of the Association for Computational Linguistics (Volume 2: Short Papers), pages 1577–1585, Toronto, Canada. Association for Computational Linguistics.
- [5] Severin Laicher, Sinan Kurtyigit, Dominik Schlechtweg, Jonas Kuhn, and Sabine Schulte im Walde. 2021. Explaining and Improving BERT Performance on Lexical Semantic Change Detection. In Proceedings of the 16th Conference of the European Chapter of the Association for Computational Linguistics: Student Research Workshop, pages 192–202, Online. Association for Computational Linguistics.
- [6] Montanelli, Stefano, and Francesco Periti. "A Survey on Contextualised Semantic Shift Detection." arXiv preprint arXiv:2304.01666 (2023).

**Paper Topic And Main Contributions:**

The authors address the problem of identifying lexical semantic changes, i.e. words that change their meaning over time. They propose an original and straightforward approach called Swapping based Semantic Change Detection (SSCD), which involves randomly swapping contexts of a target word between the two corpora and analyzing the distribution of contextualized word embeddings obtained from a pretrained BERT model. SSCD proved to be effective over the SemEval-2020 Task1 benchmark for English, German, and Swedish.

**Reasons To Accept:**

While the proposed method is relatively straightforward, its originality advances research in the field of semantic change detection. It prompts an exploration into the reliability and robustness of state-of-the-art approaches, thereby complementing the use of the permutation test to validate prediction reliability. However, although the proposed methods outperform the compared approach, they do not surpass the state of the art, as mentioned. A more in-depth discussion of the results is required.

**Reasons To Reject:**

The paper's comparison is not exhaustive, as there exist other state-of-the-art approaches that exhibit superior performance. While it is fair to exclude comparisons with approaches that involve special training or fine-tuning (which should be presented separately, e.g. [1]), it is essential to include comparisons with other approaches that use embeddings from pre-trained or fine-tuned (through continued pretraining) models. In this context, the paper does not beat the state-of-the-art methods, as indicated in the abstract (row 024), since there are other approaches that outperform it (row 184, i.e. [1]). Instead, it outperforms the compared method (row 232, 397).

When analyzing the ranking task for the SemEval English dataset, it is important to consider and mention other results, such as:

Finetuned/Trained models:
- [1] presents results indicating that the smaller the model, the better the performance. In particular, [1] reports scores of .520 with BERT-base, .589 with BERT-small, and .627 with BERT-tiny
- [2,3] report a score of .605 using ELMo
- [4] reports .757

Pretrained models:
- Other works exhibit superior results such as .571 [5]

Although SSCD achieved a slightly inferior score (.563), it cannot be argued that it outperformed the state of the art. For a comprehensive comparison across various languages, I suggest referring to [6].

Moreover, it would be beneficial to provide a more comprehensive justification and discussion regarding the use of different distances and context sampling strategies (the sampling strategies has the potential to be very interesting if further investigated). Specifically, the experiments might benefit from a more explicit explanation of the specific advantages offered by each measure, along with the underlying intuition behind the methods that achieved the most favorable result.

Other statements, such as the one in row 226 regarding the frequency of words, should be better motivated, as the impact of frequency is not discussed throughout the paper. If it refers to the potential use of pre-trained models instead of training a new model to handle low-frequent word, the reference to the previous approach by Liu et al. is not suitable, given that they also employ pre-trained models.

As a minor point:
Lastly, it would be beneficial if the authors could provide some justification for their choice to use multilingual BERT (mBERT). Typically, multilingual models are commonly used for the following reasons: i) addressing the absence of monolingual models in HuggingFace for Latin, ii) leveraging the models' cross-lingual transferability property, iii) leveraging the power of XLM-R. However, this paper does not test the Latin benchmark, nor does it exploit cross-lingual transferability or discuss the use of XLM-R. Thus, it might be worth considering to use monolingual models that may potentially yield higher results in this specific context. A comparison between mBERT and XLM-R (or monolingual BERT) may potentially provide valuable insights and enhance the overall contribution of the paper.

**Reproducibility:**

4: Could mostly reproduce the results, but there may be some variation because of sample variance or minor variations in their interpretation of the protocol or method.

**Reviewer Confidence:**

5: Positive that my evaluation is correct. I read the paper very carefully and I am very familiar with related work.

**Typos Grammar Style And Presentation Improvements:**

- row 011: occur -> occurs
- row 056: challening -> challenging
- row 089: distribitions -> distributions
- row 114: distribition -> distribution
- row 135: establisning -> establishing
- row 145: unsupervided -> unsupervised
- Table 2: unsupervied -> unsupervised
- Table 2: averagd -> averaged
- row 572: semanctic -> semantic
- row 488: the curves are not overlapping, but identical. Might there be an error? Otherwise you should explain better this behavior.
- Table 4: Is the value of .627 correct? Should it instead be .563? If this is not the case, please provide a better explanation and discussion of the value

---

> ### Author Rebuttal · Authors · 2023-08-27
>
> We appreciate your time and effort to provide insightful feedback on our paper. Below, we provide a summary of the main points and our corresponding responses.
>
> Q1: *When analyzing the ranking task for the SemEval English dataset, it is important to consider and mention other results, such as:
> Finetuned/Trained models: [4] reports .757
> Pretrained models: Other works exhibit superior results such as .571 [5]*
> A1: First, please note that [4] uses many external datasets in its model. This makes it difficult to adapt to languages that are not present in the external datasets. On the other hand, we show that our method can be adapted to any language model, and that it is sufficient to detect semantic change even in the pre-trained multilingual model.
> Second, our experiments were conducted with uniformly lemmatised data in all languages. The high score in [2] is due to using the original data, which is 0.493 when compared with the lemmatised dataset under the same conditions, and our method achieves a higher performance. As mentioned in [2], our method is expected to perform even better if it is able to accurately extract usages from original data for any language. This discussion will be added to the camera ready.
>
> Q2: *Specifically, the experiments might benefit from a more explicit explanation of the specific advantages offered by each measure, along with the underlying intuition behind the methods that achieved the most favorable result.*
> A2: As already stated in Lines 465-467 (Table 2), divergence-based measures outperform distance-based metrics for SCD. Among the distance-based metrics, Cosine and Correlation outperform the others. We believe this phenomenon is due to the fact that by swapping sentences in two time periods, the set of vectors after the swap becomes sparse, so that the average part of the set alone cannot be used to properly calculate the similarity of the swapped sentences. We also consider that this is the same as the example of “cell” in Aida+Bollegala [ACL findings 2023], where the set of vector increases by acquiring a new meaning of “cell phone“, while retaining the old meanings of “prison” and “biology,” and the mean component of the vector set alone cannot detect the semantic change. Due to the 8 page limit, we could not include the above details in the paper, but with the extra 1 page allowed, we can include this discussion in the camera ready paper if accepted.
>
> Q3: *row 226 regarding the frequency of words, should be better motivated, as the impact of frequency is not discussed throughout the paper.*
> A3: In the method of Liu et al. [Eval4NLP 2021], low-frequency words are removed from the prediction results. In order to make appropriate predictions for infrequent words, we use context swapping to calculate the degree of semantic change. In experiments, on benchmarks with words with a frequency of less than 10 times, such as Liverpool FC, we achieved comparable performance on the pre-trained MLM against the finetuned MLM of Liu et al. [Eval4NLP 2021].
>
> Q4: *A comparison between mBERT and XLM-R (or monolingual BERT) may potentially provide valuable insights and enhance the overall contribution of the paper.*
> A4: By using a multilingual MLM we can use the same model for multiple languages, without having to train separate models for each language. To investigate the feasibility of this approach, we used mBERT as the base MLM in these evaluations.

---

### Official Review · Reviewer_8L1o · 2023-08-10

**Soundness:** 2

**Excitement:**

3: Ambivalent: It has merits (e.g., it reports state-of-the-art results, the idea is nice), but there are key weaknesses (e.g., it describes incremental work), and it can significantly benefit from another round of revision. However, I won't object to accepting it if my co-reviewers champion it.

**Missing References:**

[1] Haim Dubossarsky, Daphna Weinshall, and Eitan Grossman. 2017. Outta Control: Laws of Semantic Change and Inherent Biases in Word Representation Models. In Proceedings of the 2017 Conference on Empirical Methods in Natural Language Processing, pages 1136–1145, Copenhagen, Denmark. Association for Computational Linguistics.

[2] Severin Laicher, Sinan Kurtyigit, Dominik Schlechtweg, Jonas Kuhn, and Sabine Schulte im Walde. 2021. Explaining and Improving BERT Performance on Lexical Semantic Change Detection. In Proceedings of the 16th Conference of the European Chapter of the Association for Computational Linguistics: Student Research Workshop, pages 192–202, Online. Association for Computational Linguistics.

[3] Pierluigi Cassotti, Lucia Siciliani, Marco DeGemmis, Giovanni Semeraro, and Pierpaolo Basile. 2023. XL-LEXEME: WiC Pretrained Model for Cross-Lingual LEXical sEMantic changE. In Proceedings of the 61st Annual Meeting of the Association for Computational Linguistics (Volume 2: Short Papers), pages 1577–1585, Toronto, Canada. Association for Computational Linguistics.

[4] Francesco Periti, Alfio Ferrara, Stefano Montanelli, and Martin Ruskov. 2022. What is Done is Done: an Incremental Approach to Semantic Shift Detection. In Proceedings of the 3rd Workshop on Computational Approaches to Historical Language Change, pages 33–43, Dublin, Ireland. Association for Computational Linguistics.

**Paper Topic And Main Contributions:**

The presented research introduces Swapping-based Semantic Change Detection (SSCD), an unsupervised method designed to predict shifts in word meaning between distinct text corpora. By swapping contexts between two corpora and analyzing the distribution of contextualized word embeddings from a pre-trained masked language model (MLM), SSCD determines whether a target word's semantics change.

**Questions For The Authors:**

A) Why have you excluded Latin (SemEval 2020 Task 1) from the experiments?

B) Why did you not consider Mahalanobis distance which is usually used to compare Gaussian distributions?

**Reasons To Accept:**

This paper offers a compelling contribution to the field of lexical semantic change detection while drawing parallels to the work proposed by [1], introducing an element of innovation to the field and bringing forth a new way to evaluate the significance of detected semantic shifts.

**Reasons To Reject:**

Due to several significant concerns, the paper's experimental setting needs more convincing evidence to support its claims. While the proposed approach is independent of specific models or measures of lexical semantic change, the experiments disproportionately focus on distance and divergence metrics. It is essential to assess the approach independently of specific models to gauge its potential to enhance or diminish the performance of state-of-the-art (SOTA) models such as [2] or [3]. Contrary to the authors' assertion of achieving SOTA results for English, their comparison with [2] under the same setting without the SSCD approach yields a result of 0.571 [4]. This incongruity raises questions about the reliability of the proposed method's effectiveness, underscoring the need for a more comprehensive and accurate experimental design.

**Reproducibility:**

4: Could mostly reproduce the results, but there may be some variation because of sample variance or minor variations in their interpretation of the protocol or method.

**Reviewer Confidence:**

4: Quite sure. I tried to check the important points carefully. It's unlikely, though conceivable, that I missed something that should affect my ratings.

---

> ### Author Rebuttal · Authors · 2023-08-27
>
> We appreciate your time and effort to provide insightful feedback on our paper. Below, we provide a summary of the main points and our corresponding responses.
>
> Q1: *the experiments disproportionately focus on distance and divergence metrics*
> A1: In this paper, we do not propose distance/divergence metrics, but simply use in the experiments what has already been proposed in previous studies. This imbalance arises because the previous studies happened to use more distance metrics compared to divergence measures.
>
> Q2: *Contrary to the authors' assertion of achieving SOTA results for English, their comparison with [2] under the same setting without the SSCD approach yields a result of 0.571 [4].*
> A2: Our experiments were conducted with uniformly lemmatised data in all languages. The high score in [2] is due to using the original data, which is 0.493 when compared with the lemmatised dataset under the same conditions, and our method achieves a higher performance. As mentioned in [2], our method is expected to perform even better if it is able to accurately extract usages from original data for any language, taking into account their utilisation. We will include this discussion in the camera ready version, if accepted.
>
> Q3: *Why have you excluded Latin (SemEval 2020 Task 1) from the experiments?*
> A3: In our evaluations, we focused on languages that are currently being actively used.
>
> Q4: *Why did you not consider Mahalanobis distance which is usually used to compare Gaussian distributions?*
> A4: Please note that our proposed swapping-based semantic change detection method is independent of any particular distance function. Our main contribution of this paper is to propose and analyse the swapping method, and we do not propose any distance/divergence metrics. Instead, we follow Aida+Bollegala [ACL findings 2023] for the distance/divergence measures. As for the specific question about Mahalanobis distance, it can also be used as a distance measure with our proposed SSCD. However, please note that we approximate the covariance matrices of Gaussians to be diagonal in our experiments, which makes Mahalanobis distance somewhat similar to the Euclidean in this case.

---

### Meta-Review · Area_Chair_JMXM · 2023-09-19

**Recommendation:** 3

**Metareview:**

The methodology is fairly clear and appears to offer interesting results for the intended taks. The linguistic or cognitive significance are however not explicitly highlighted.

---

### Decision · Program_Chairs · 2023-10-07

**Decision:**

Accept-Findings

**Comment:**

The methodology is fairly clear and appears to offer interesting results for the intended taks. The linguistic or cognitive significance are however not explicitly highlighted.